# Pathophysiological Potentials of NRF3-Regulated Transcriptional Axes in Protein and Lipid Homeostasis

**DOI:** 10.3390/ijms222312686

**Published:** 2021-11-24

**Authors:** Tsuyoshi Waku, Akira Kobayashi

**Affiliations:** 1Laboratory for Genetic Code, Department of Medical Life Systems, Faculty of Life and Medical Sciences, Doshisha University, Kyotanabe 610-0394, Japan; twaku@mail.doshisha.ac.jp; 2Laboratory for Genetic Code, Graduate School of Life and Medical Sciences, Doshisha University, Kyotanabe 610-0394, Japan

**Keywords:** NRF3, protein homeostasis, lipid homeostasis, proteasome, translation, GGPP, macropinocytosis, cancer, obesity

## Abstract

NRF3 (NFE2L3) belongs to the CNC-basic leucine zipper transcription factor family. An NRF3 homolog, NRF1 (NFE2L1), induces the expression of proteasome-related genes in response to proteasome inhibition. Another homolog, NRF2 (NFE2L2), induces the expression of genes related to antioxidant responses and encodes metabolic enzymes in response to oxidative stress. Dysfunction of each homolog causes several diseases, such as neurodegenerative diseases and cancer development. However, NRF3 target genes and their biological roles remain unknown. This review summarizes our recent reports that showed NRF3-regulated transcriptional axes for protein and lipid homeostasis. NRF3 induces the gene expression of *POMP* for 20S proteasome assembly and *CPEB3* for NRF1 translational repression, inhibiting tumor suppression responses, including cell-cycle arrest and apoptosis, with resistance to a proteasome inhibitor anticancer agent bortezomib. NRF3 also promotes mevalonate biosynthesis by inducing *SREBP2* and *HMGCR* gene expression, and reduces the intracellular levels of neural fatty acids by inducing *GGPS1* gene expression. In parallel, NRF3 induces macropinocytosis for cholesterol uptake by inducing *RAB5* gene expression. Finally, this review mentions not only the pathophysiological aspects of these NRF3-regulated axes for cancer cell growth and anti-obesity potential but also their possible role in obesity-induced cancer development.

## 1. Introduction

Protein and lipid homeostasis is crucial for cell survival and proliferation, and the defects interfere with several diseases, such as neurodegeneration, cancer development, metabolic disorder, and obesity [1,2,3,4]. NRF3 (nuclear factor erythroid 2-like 3; NFE2L3) belongs to the cap‘n’collar (CNC)-basic leucine zipper transcription factor family, and has two homologs: NRF1 (nuclear factor erythroid 2-like 1; NFE2L1) and NRF2 (nuclear factor erythroid 2-like 2; NFE2L2) [5,6]. NRF1 mainly maintains the proteasome activity by comprehensively inducing the expression of most proteasome-related genes [7]. *Nrf1*-null mice suffer from embryonic lethality [8]. Thus, neuron, liver, or osteoblast-specific *Nrf1* knockout mice have been generated and show tissue defects, such as neurodegeneration, nonalcoholic steatohepatitis, and bone loss [9,10,11,12]. NRF2 is crucial for the cytoprotective mechanisms against xenobiotic and oxidative stress [13]. NRF2 also activates genes encoding enzymes for glutaminolysis, shifting the metabolic flux of glutamine to the glutathione synthesis pathway [14,15]. *Nrf2*-null mice do not respond to oxidative stress [16], whereas *Kelch-like ECH-associated protein 1* (*Keap1*)-null mice demonstrate postnatal lethality by the constitutive activation of *Nrf2* [17]. Meanwhile, the biological function of NRF3 has long remained unclear because *Nrf3*-null mice develop and grow normally under physiological conditions [18,19,20]. However, studies on NRF3 have recently increased [21,22,23]. For example, we reported that the *NRF3* gene is highly expressed in several cancers [24].

NRF3 and NRF1 proteins are anchored to the endoplasmic reticulum (ER), and are degraded through ER-associated degradation. Proteotoxic stress, such as proteasome inhibition, leads to the cleavage of these proteins by the aspartic protease DNA damage-inducible 1 homolog 2 (DDI2), resulting in the nuclear translocation of cleaved NRF3 and NRF1 proteins for transcription activation [25,26,27]. Meanwhile, NRF2 proteins are negatively regulated by a cytosolic E3 ligase adaptor protein KEAP1, and are activated in response to oxidative stress [28]. In the nucleus, activated NRF proteins heterodimerize with small musculoaponeurotic fibrosarcoma (sMAF) proteins, including MAFF, MAFG, and MAFK, and bind to a consensus sequence called antioxidant response element (ARE; TGA[G/C]NNNGC) [29,30]. To date, we have identified several NRF3 target genes that coordinate protein and lipid homeostasis by gene expression analysis based on DNA microarray, real-time quantitative PCR, and chromatin immunoprecipitation (ChIP) experiments.

This review first introduces that NRF3 promotes cancer development through proteasome regulation, by inducing the gene regulation of *proteasome maturation protein* (*POMP*) [31] and *cytoplasmic polyadenylation element-binding protein 3* (*CPEB3*) [32]. Then, this review describes the gene expression network of NRF3-regulated lipid metabolism, including *sterol regulatory element-binding protein 2* (*SREBP2*) and *hydroxy-methylglutaryl-CoA reductase* (*HMGCR*) [33]. NRF3 also induces the gene expression of *GGP synthase 1* (*GGPS1*) for geranylgeranyl pyrophosphate (GGPP)-mediated lipogenesis inhibition and *ras-related small GTPase protein* (*RAB5*) for macropinocytic cholesterol uptake [33]. Finally, this review remarks on the pathophysiological potential of these NRF3-regulated axes for cancer and obesity.

## 2. Assembly of the Ubiquitin-Independent 20S Proteasome

### 2.1. POMP, a 20S Proteasome Assembly Factor

The 26S proteasome is essential for ubiquitin-dependent protein degradation and consists of two subcomplexes: a 20S proteasome and a 19S-regulatory particle (RP) [34]. Several chaperones strictly coordinate the assembly of 20S proteasome and 19S-RP [35,36]. NRF1 induces the expression of almost all proteasome-related genes required for 26S proteasome [7]. Meanwhile, NRF3 does not affect the expression of almost all proteasome-related genes, but induces the gene expression of *POMP* [31], a chaperone of the 20S proteasome assembly [37]. ChIP experiments showed an ARE-like sequence (TGAGCGGCG) near the transcription start site of the *POMP* gene as the NRF3 binding region [31] (Figure 1A). Furthermore, *POMP*-ARE mutations using CRISPR/Cas9-based genome editing reduced not only NRF3 recruitment on *POMP*-ARE but also *POMP* gene expression induced by NRF3 [31]. Proteasome activity assays using a fluorogenic substrate showed that NRF3 increases the amount and activity of 20S proteasome [31] (Figure 1B). These results provided direct evidence that *POMP* is an NRF3 target gene for enhancing the 20S proteasome activity.

### 2.2. NRF3-POMP-20S Proteasome Assembly Axis for Cancer Development

The 20S proteasome, a homodimer of a half-mer proteasome composed of an outer α-ring and an inner β-ring, contains proteolytic sites with different specificities: chymotrypsin-, caspase-, and trypsin-like activities. Meanwhile, the 20S proteasome lacks the 19S-RP that selects and unfolds ubiquitin substrates. Previous studies suggested that the 20S proteasome contributes to the ubiquitin-independent degradation of several tumor suppressor proteins, such as p53 and retinoblastoma (Rb) [38]. Surprisingly, NRF3 decreases p53 and Rb proteins without alteration of their mRNA levels under treatment with a ubiquitin-activating enzyme E1 inhibitor TAK-243, which inhibits 26S proteasome-mediated protein degradation by covalently binding with ubiquitin proteins [39] (Figure 1B). Furthermore, *POMP*-ARE mutation impairs the NRF3-mediated reduction in p53 and Rb protein, irrespective of TAK-243 treatment. More importantly, p53 and Rb inhibit cancer cell proliferation by inducing cell-cycle arrest or apoptosis in response to DNA damage [40]. NRF3 suppresses the expression of p53 target genes, including the cell-cycle inhibitory effector gene *p21* [41] and the proapoptotic gene *PUMA* (*p53 upregulated modulator of apoptosis*) [42]. NRF3 further inhibits p53-dependent cell-cycle arrest and apoptosis induction, leading to continuous cancer cell growth [31] (Figure 1B). These results indicated that the NRF3-POMP-20S proteasome assembly axis affects the ubiquitin-independent degradation of endogenous p53 and Rb proteins.

The proteasome is a target for cancer chemotherapy, and several proteasome inhibitors have been developed as anticancer agents. Among proteasome inhibitor anticancer agents, bortezomib (BTZ) inhibits both 20S and 26S proteolytic activities by binding to catalytic sites within the 20S proteasome [43]. Expectedly, upregulation of the NRF3-POMP-20S proteasome axis confers resistance to BTZ [31] (Figure 1B). Furthermore, xenograft and hepatic metastatic mouse models showed that NRF3 increases tumorigenesis and metastasis, whereas *POMP*-ARE mutation inhibits this tumor burden [31]. More importantly, clinical analyses indicated a negative correlation between *POMP/NRF3* mRNA levels and the survival rates of patients with colorectal adenocarcinoma, where *NRF3* is highly expressed [31]. These insights shed light on the crucial function of the NRF3-POMP-20S proteasome axis on cancer development, by inhibiting tumor suppression signals of p53 and Rb through ubiquitin-independent degradation. The upregulation of the axis also confers resistance to a BTZ-type proteasome inhibitor [44].

## 3. Complementary Maintenance of Proteasome with NRF1

### 3.1. CPEB3, a Translational Repressor of NRF1

NRF3 increases 20S proteasome activity through *POMP* expression [31], whereas NRF1 maintains 26S proteasome activity by inducing the expression of almost all proteasome-related genes under proteasome inhibition [7], implying the biological relevance of NRF1 and NRF3 for proteasome activity. In fact, the double knockdown of NRF1 and NRF3 impairs basal proteasome activity in living cells [32]. Compared to the single knockdown of NRF1 or NRF3, double knockdown of NRF1 and NRF3 reduces several proteasome-related genes, including *PSMB3*, *PSMB7*, *PSMC2*, *PSMD3*, *PSMG2*, *PSMG3*, and *POMP* [32]. ChIP experiments showed an ARE sequence near the transcription start site of each gene. These results indicated that NRF1 and NRF3 complementarily induce the expression of several proteasome-related genes to maintain the proteasome activity [32].

Interestingly, NRF3 represses the translation of NRF1 proteins by decreasing the amount of *NRF1* mRNA in polysomes, although NRF3 does not only affect the levels of *NRF1* mRNA but also the degradation of NRF1 proteins [32]. Gene expression analysis identified *CPEB3* as the candidate NRF3 target gene for this NRF1 translation repression [32] (Figure 2A). CPEB family proteins recognize a CPEB recognition motif (5′-UUUUA-3′, CPE) in the 3′-untranslated region (UTR) of a target gene for translation regulation [45]. CPEB3 interacts with the *NRF1*–3′-UTR which contains five CPEs, decreasing NRF1 protein levels and the amount of *NRF1* mRNA in polysomes [32] (Figure 2A). Meanwhile, NRF3 deficiency or CPE mutation of the *NRF1*–3′-UTR increases NRF1 translation [32] (Figure 2B). These results indicate that NRF3 directly induces *CPEB3* gene expression, and then CPEB3 inhibits ribosome recruitment to *NRF1* mRNA, resulting in the repression of NRF1 translation.

### 3.2. Clinical Significance of the NRF3-CPEB3-NRF1 Translational Repression Axis

In NRF3-deficienct cells, CPEB3 represses NRF1 translation and then reduces the expression levels of *PSMB3*, *PSMB7*, *PSMC2*, *PSMG2*, and *POMP* genes, resulting in the suppression of 26S proteasome activity [32]. CPEB3 also confers resistance to BTZ in NRF3-deficient cells [32]. Furthermore, colorectal cancer patients with higher *CPEB3*/*NRF3*-expressing tumors exhibited shorter overall survival rates, but higher *CPEB3*/*NRF1* expression was not associated with poor prognosis [32]. These results suggested that the NRF3-CPEB3-NRF1 translational repression axis is involved in cancer development by shunting ubiquitin-dependent protein degradation through the NRF1–26S proteasome regulatory axis to ubiquitin-independent protein degradation through the POMP-20S proteasome axis (Figure 2).

## 4. Reprogramming of Lipid Metabolism

### 4.1. NRF3-SREBP2-HMGCR Axis for Mevalonate Biosynthesis

Lipids, such as cholesterol and fatty acids, influence cell signaling, energy storage, and membrane formation. SREBPs are membrane-bound transcription factors crucial for lipid metabolism [46]. In response to cholesterol depletion, SREBP1 and SREBP2 proteins are cleaved in the Golgi apparatus, resulting in the translocation to the nucleus. SREBP1 induces the gene expression of enzymes required for fatty acid biosynthesis and adipocyte differentiation, whereas SREBP2 induces the gene expression of enzymes required for mevalonate/cholesterol biosynthesis.

NRF3 induces the expression of several SREBP2 target genes, such as *hydroxy-methylglutaryl-CoA synthase 1* (*HMGCS1*) and *HMGCR*, encoding a rate-limiting enzyme in mevalonate/cholesterol biosynthesis [47] (Figure 3(1)). ChIP experiments with previously published ChIP sequencing data [48] indicated that NRF3 binds to both AREs in *SREBP2* and *HMGCR* promoters, and that SREBP2 binds to the site nearby *HMGCR*-ARE [33]. Moreover, NRF3 interacts with the active form of SREBP2 [33] (Figure 3(1)), implying that NRF3 and SREBP2 form a transcriptional complex for *HMGCR* gene expression. Luciferase reporter assays containing both ARE and SREBP2 binding sites showed a synergistic transcriptional activity of NRF3 and SREBP2 through the *HMGCR* promoter [33]. Taken together, NRF3 promotes mevalonate biosynthesis by upregulating the SREBP2-HMGCR axis (Figure 3(1)).

### 4.2. NRF3-GGPS1-GGPP Production Axis for Lipogenesis Inhibition

Interestingly, NRF3 does not affect the intracellular levels of cholesterol, even if NRF3 increases the expression levels and enzymatic activity of HMGCR. Meanwhile, NRF3 reduces that of lanosterol [33]. Lanosterol is not only a precursor of cholesterol but also a downstream metabolite of farnesyl pyrophosphate, which is also metabolized to GGPP in a reaction catalyzed by GGPS1 (Figure 3(2)). Furthermore, NRF3 directly induces *GGPS1* expression [33], implying that NRF3 reprograms cholesterol biogenesis to the production of GGPP rather than lanosterol. GGPP suppresses SREBP1-dependent fatty acid biosynthesis and intracellular lipid accumulation [49,50]. In fact, DNA microarray analysis showed a negative correlation between the expression levels of *NRF3* and genes related to fatty acid metabolism [33]. More directly, intracellular levels of neutral lipids are increased by NRF3 knockdown and reduced by GGPP treatment [33] (Figure 3(2)). Consistently, a few body mass index-associated genomic loci near the *NRF3* gene have been identified previously [51,52]. These results indicated the potential role of the NRF3-GGPS1-GGPP production axis (Figure 3(2)).

### 4.3. NRF3-RAB5-Macropincytosis Induction Axis for Cholesterol Uptake

Intracellular cholesterol is derived from not only de novo biosynthesis but also endocytic uptake [53]. NRF3 decreases lanosterol levels, but it does not change cholesterol levels in cells [33] (Figure 3(3)), implying that NRF3 enhances endocytosis for cholesterol uptake to compensate for the potential depletion in cholesterol levels following lanosterol reduction. Low-density lipoprotein receptor (LDLR) is a key endocytosis regulator of LDL [54]. However, NRF3 does not induce *LDLR* gene expression. Meanwhile, NRF3 induces the gene expression of three isoforms of *RAB5A*, *RAB5B*, and *RAB5C* [33] (Figure 3(3)). These RAB5 proteins act as early endocytosis regulators [55] and are involved in macropinocytosis, a bulk and fluid-phase endocytosis process [56]. NRF3 further increases posttranslational prenylation RAB5 proteins [33] essential for proper localization and function in membranes [57]. The previous section (Figure 3(2)) described that NRF3 induces the production of GGPP, which functions as a required substrate for protein prenylation [58]. Altogether, NRF3 enhances RAB5-mediated endocytosis rather than LDLR-mediated endocytosis for cholesterol uptake through GGPP production (Figure 3, (3)). NRF3 enhances the uptake of fluorogenic LDL in a RAB5-dependent manner. Moreover, NRF3-enhanced uptake of other fluorogenic macropinocytosis indicators based on 70 kDa dextran and bovine serum albumin is abolished by treatment with 5-(*N*-ethyl-*N*-isopropyl)amiloride [33], also known as an inhibitor of macropinocytosis and a selective blocker of Na^+^/H^+^ exchanger [59]. Similarly, NRF3 enhances the uptake of two fluorogenic cholesterols through macropinocytosis [33]. These results indicated the crucial function of the NRF3-RAB5-macropinocysosis induction (NRF3-RAB5-macropinocysosis) axis on cholesterol uptake (Figure 3(3)). The next section discusses the pathophysiological potential of this axis.

## 5. Concluding Remarks

Increased ubiquitin-independent proteasomal activity causes tumor growth, metastasis, and resistance to the proteasome inhibitor BTZ. NRF3 induces *POMP* expression, leading to ubiquitin-independent protein degradation of tumor suppressors Rb and p53 (Figure 1). Upregulation of the POMP-20S proteasome axis further results in poor prognosis of colorectal cancer patients [31]. NRF3 also induces the expression of proteasome-related genes in parallel with NRF1 translational repression by inducing *CPEB3* expression [32] (Figure 2A). If the *NRF3* gene is deficient, NRF1 escapes from CPEB3-mediated translational repression, and complementarily plays a transcriptional role for the robust maintenance of basal proteasome activity in cancer cells (Figure 2B). Although NRF3 shares several target genes with NRF1 and NRF2 on ARE [30,60], this review showed the translation-mediated crosstalk between NRF3 and NRF1.

*Nrf1* is ubiquitously expressed in normal tissues, and *Nrf1* knockout mice suffer from embryonic lethality [8]. Meanwhile, *Nrf3* expression levels are low, except in several mouse tissues, such as the placenta [5,6], and *Nrf3* knockout mice do not exhibit any obvious abnormalities under normal physiological conditions [18,19,20]. However, *NRF3* is highly expressed in many cancer cells [31], implying that the proteasome in cancer or normal cells is maintained through the CPEB3-NRF1 axis or the negative feedback regulation of NRF1. Higher *CPEB3*/*NRF3* expression, but not higher *CPEB3*/*NRF1* expression, is associated with poor prognosis of cancer patients [32]. Therefore, NRF1 maintains a 26S proteasome activity for normal development, whereas NRF3 alternatively maintains 20S proteasome activity for cancer development through both POMP-20S proteasome and CPEB3-NRF1 axes.

Furthermore, NRF3 is involved in lipid metabolism through three regulatory axes [33] (Figure 3): (1) NRF3 induces the gene expression of *SREBP2* required for cholesterol biosynthesis through the mevalonate pathway. NRF3 also leads to SREBP2 activation through direct induction of gene expression. NRF3 and SREBP2 synergistically induce *HMGCR* expression and the following mevalonate biosynthesis. (2) NRF3 then upregulates GGPS1-mediated GGPP production for lipogenesis inhibition. (3) In parallel, NRF3 confers RAB5-mediated induction of macropinocytosis for cholesterol uptake. This gene expression is induced in colon and/or rectal tissue of newly generated NRF3-transgenic mice [33]. Dietary cholesterol in the blood is absorbed in the intestine [61], and its dysregulation is associated with obesity, resulting in an increased risk of cardiovascular diseases (CVD) and colorectal cancer [62,63]. The gut microbiota has been identified as a CVD risk factor and regulates host cholesterol homeostasis [64,65,66], suggesting the pathophysiological potential of the NRF3-regulated host lipid metabolism to the gut–heart connection through the gut microbiota.

Epidemiological studies have associated obesity with a range of cancers [67,68]. Furthermore, many efforts have been made to identify the key factor for obesity-induced cancer, including insulin resistance, increased steroid hormones and adipokine, and aberrant inflammation [69]. However, these findings implied possible opposite roles for NRF3 in obesity-induced cancer development (Figure 4): NRF3-SREBP2-HMGCR and the following NRF3-GGPS1-GGPS axes regulate lipid homeostasis and confer resistance to obesity through lipogenesis inhibition, while the NRF3-POMP-20S proteasome and NRF3-CPEB3-NRF1 axes regulate protein homeostasis and confer ubiquitin-independent protein degradation for continuous cancer cell growth. This review further showed that NRF3 maintains cholesterol homeostasis through the RAB5-macropinocytosis axis (Figure 3, (3)). Interestingly, macropinocytosis is associated with obesity-related disorders, such as increased diabetic mouse macrophages and chronic inflammation [70,71]. Furthermore, NRF2 induces macropinocytosis and contributes to the escape of autophagy-deficient cancer cells from metabolic decline and anticancer drugs, such as gemcitabine and doxorubicin, which target the anabolic dependencies of cancer cells [72,73]. These insights implied the possibility that the NRF3-RAB5-macropinocytosis axis paradoxically interferes with obesity-induced cancer development through attenuation of obesity-induced inflammation and resistance to therapy targeting cancer anabolism (Figure 4).

A big issue of the NRF3 study is identifying the endogenous cue of NRF3 activation, although NRF3 is experimentally activated by treatment with a proteasome inhibitor. Recently, NRF1 senses cholesterol levels in the ER membrane through the cholesterol recognition amino acid consensus motif domain (CRAC), and it is activated in response to cholesterol depletion [74]. Because the CRAC domain is conserved in NRF3 proteins [27], NRF3 acts as a cholesterol sensor in the ER membrane similarly to NRF1.

## Figures and Tables

**Figure 1 ijms-22-12686-f001:**
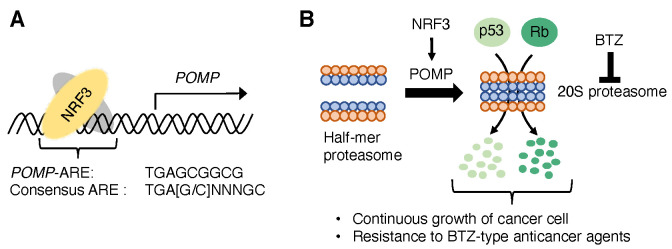
NRF3-POMP-20S proteasome assembly axis. (**A**) NRF3 directly induces *POMP* expression by binding to *POMP*-ARE, which is slightly different from consensus ARE. (**B**) Upregulation of the NRF3-POMP axis enhances a dimerization of a half-mer proteasome (known as the 20S proteasome assembly). The increased 20S proteasome confers the ubiquitin-independent degradation of p53 and Rb proteins, resulting in the rapid and continuous growth of cancer cells. Aberrant upregulation of the axis also confers resistance to a BTZ-type anticancer agent.

**Figure 2 ijms-22-12686-f002:**
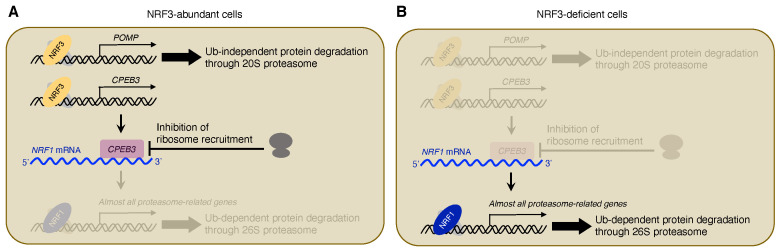
NRF3-CPEB3-NRF1 translational repression axis. (**A**) In NRF3-abundant cells, NRF3 directly induces *CPEB3* expression, resulting in the repression of NRF1 translation. In parallel, NRF3 confers ubiquitin (Ub)-independent protein degradation through the POMP-20S proteasome axis. (**B**) In NRF3-deficient cells, NRF1 escapes from CPEB3-mediated translational repression and confers ubiquitin (Ub)-dependent protein degradation.

**Figure 3 ijms-22-12686-f003:**
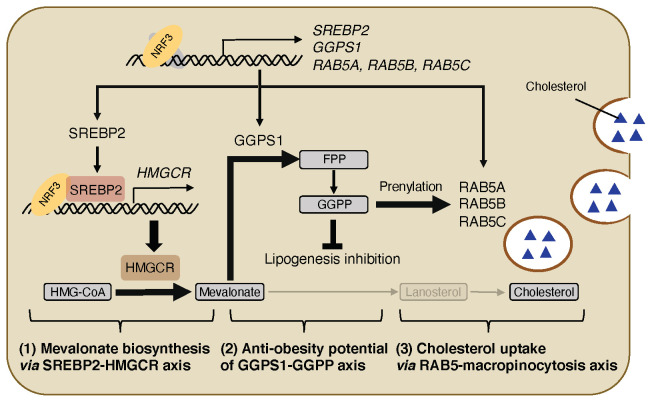
NRF3-regulated lipid metabolism through three axes. (**1**) NRF3 activates *SREBP2* by directly inducing gene expression. NRF3 and SREBP2 synergistically induce *HMGCR* gene expression, promoting mevalonate biosynthesis. (**2**) In parallel, NRF3 induces *GGPS1* expression and then reprograms cholesterol biosynthesis to GGPP production, resulting in lipogenesis inhibition. (**3**) NRF3 also induces the gene expression of three *RAB5* isoforms, resulting in cholesterol uptake through macropinocytosis.

**Figure 4 ijms-22-12686-f004:**
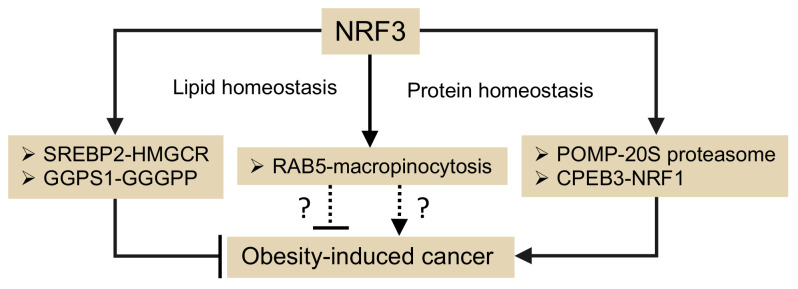
Possible roles for NRF3 in obesity-induced cancer development.

## Data Availability

Not applicable.

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
