# Peer review of "Pathophysiological Potentials of NRF3-Regulated Transcriptional Axes in Protein and Lipid Homeostasis"

_ijms, 2021, doi:10.3390/ijms222312686_

Round 1

Reviewer 1 Report

In the present manuscript, the authors reviewed the latest findings about the pathophysiological role of NRF3-regulated transcriptional axes in protein and lipid homeostasis in cancer and obesity development.

The manuscript will be attractive for readers. The manuscript is very comprehensive. The authors included many useful diagrams showing the contribution of NRF3 in the regulation of transcriptional axes in protein and lipid metabolism. The content of the manuscript is adequate to the title of a review. After checking of overlapping proposal manuscript with other papers, I have not found any significant overlap with the others reviews e.g., https://doi.org/10.3390/cancers12092681.

I have only a one minor comment to authors:

I think that in the manuscript the authors used definitely too many citations – 109! Please note that it is a short review manuscript, so the authors should describe the most important and the newest scientific work about NRF3 and its role in protein and lipid homeostasis. Please reduce the number of citations up to 80.

Author Response

I think that in the manuscript the authors used definitely too many citations – 109! Please note that it is a short review manuscript, so the authors should describe the most important and the newest scientific work about NRF3 and its role in protein and lipid homeostasis. Please reduce the number of citations up to 80.

Response: We would like to thank you for appreciating our manuscript. As per your suggestion, we reduced the number of citations up to 74.

Reviewer 2 Report

This is a timely review about the role of NRF3 in cellular homeostasis. The three aspects, proteasome, macropinocytosis, and cholesterol production are discussed separately, but I wonder what the links might be. It would be valuable to increase the conclusion section to look at some more links.

Other comments:

"NRF2 establishes a reduced intracellular redox state by inducing the expression of genes encoding phase II detoxification and antioxidant enzymes" Perhaps rephrase here: The reduced intracellular redox state is first and foremost established by glutathione. NRF2 rather mediates a stress response.

Fig 2 Both panels appear to be the same. The downward arrows should be smaller and the right arrow bolder or similar. Fig. 2B is not mentioned in the article.

Are systemic cholesterol levels elevated in NRF3 ko mice?

Author Response

This is a timely review about the role of NRF3 in cellular homeostasis. The three aspects, proteasome, macropinocytosis, and cholesterol production are discussed separately, but I wonder what the links might be. It would be valuable to increase the conclusion section to look at some more links.

Response: We would like to thank you for appreciating our manuscript. Unfortunately, we don’t have any data and idea for the relationships between proteasome, macropinocytosis, and cholesterol production. Therefore, we will address this issue in the future.

Other comments:

"NRF2 establishes a reduced intracellular redox state by inducing the expression of genes encoding phase II detoxification and antioxidant enzymes" Perhaps rephrase here: The reduced intracellular redox state is first and foremost established by glutathione. NRF2 rather mediates a stress response.

Response: We changed the sentence as follows; “NRF2 is crucial for the cytoprotective mechanisms against xenobiotic and oxidative stress”.

Fig 2 Both panels appear to be the same. The downward arrows should be smaller and the right arrow bolder or similar. Fig. 2B is not mentioned in the article.

Response: As the reviewer’s suggestion, we improved Fig.2. We also mentioned Fig.2B in the article.

Are systemic cholesterol levels elevated in NRF3 ko mice?

Response: Thank you for your important suggestion. Although we have not investigated whether the cholesterol levels elevated in NRF3 ko mice, we will address this issue in the future.